# MPC for humanoid control

- Original paper: Koji Ishihara, Takeshi D. Itoh, Jun Morimoto, Full-body optimal control toward versatile and agile behaviors in a humanoid robot, IEEE Robotics and Automation Letters, 5 (1), 2019, pp. 119-126, https://ieeexplore.ieee.org/document/8865637.
- Retrospective written by: Koji Ishihara and Jun Morimoto

## Paper TL;DR

Future humanoid robots are expected to work in real environments such as extremely hazardous situations instead of humans. However, they still have difficulty in generating human-like behaviors, especially in terms of versatility and agility.

To cope with this problem, we developed a Model Predictive Control (MPC) approach that takes full-body dynamics into account. MPC with full-body dynamics is a good candidate for generating a wide variety of agile movements because

- We do not have to design the low-level details of each movement. Such labor-intensive tasks are automated through numerical optimization.
- Unlike using a highly reduced model of a humanoid robot, the optimization process under the constraints of full-body dynamics does not restrict the generable agile movements.

A well-known control approach is a hierarchical control architecture using an inverted pendulum and inverse dynamics control. However, an optimization process under such reduced model restricts the generable agile movements since the inverted pendulum model cannot take detailed limb movements into account.

We evaluated our approach in skating tasks with simulated and real lower-body humanoids that have rollers on the feet. We are interested in generating wide variety of human-like agile motions. Then, we selected the skating movement generation as an illustrative task. As shown in the figures below, our simulated robot generated various agile motions such as flipping down from a cliff, and our real lower-body humanoid also successfully generated a movement down a slope. Details of our proposed approach are describe in our paper [1]. Our previous and preliminary study was presented in [2].

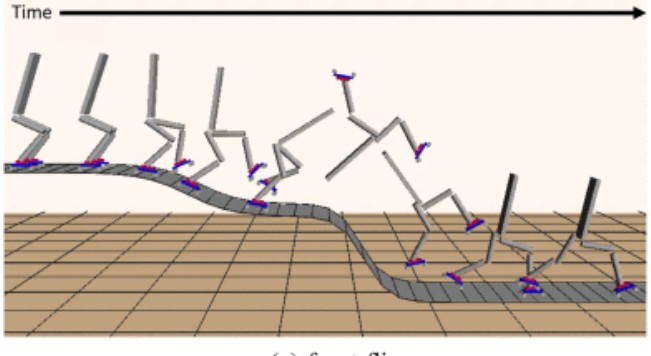

(a) front flip

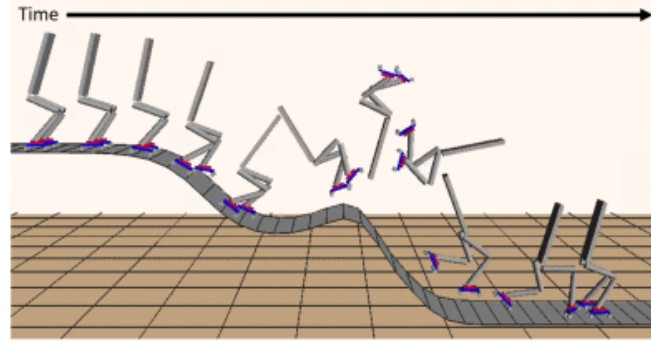

(b) back flip

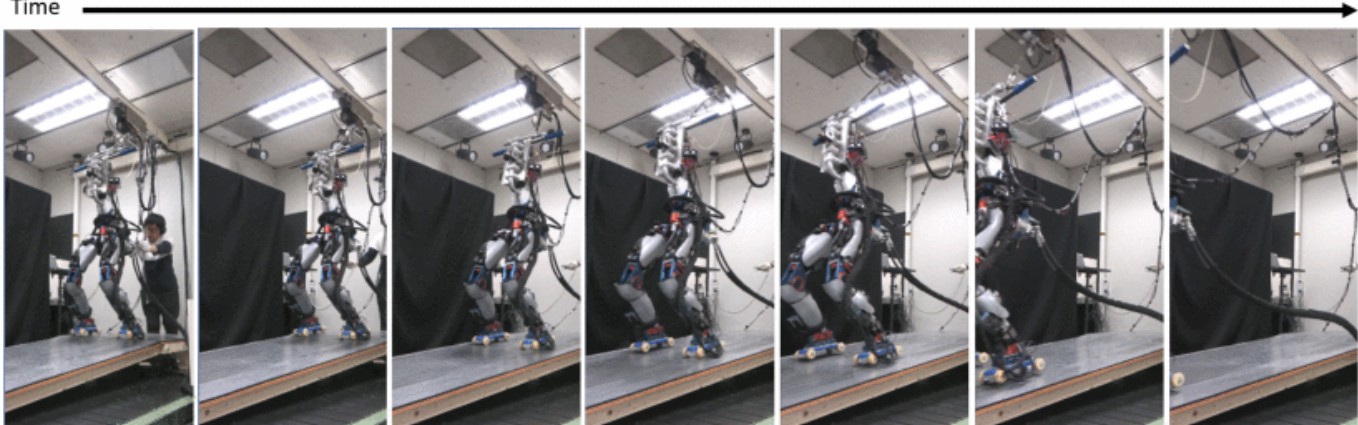

## Overall Outlook

A real-time MPC for a humanoid robot has been deemed impractical because MPC is computationally intensive. A large optimization problem needs to be solved within a short control period of the robot when full-body dynamics are used as constraints. The large computational burden of MPC often becomes a problem for a large-scale system. We dealt with this general problem by developing a computationally efficient optimization process for a humanoid robot in the paper. In the conventional MPC, full-body dynamics are optimized with fine-time resolution, and this process is time-consuming. Our proposed MPC, on the other hand, optimizes only low-dimensional fast dynamics of a humanoid robot with fine-time resolution. Other dynamics were optimized coarsely. We examined which part of bodies of a humanoid robot were able to move fast, and extracted their dynamics as the low-dimensional fast dynamics.

However, through our presentations, we rather have received questions about implementation details such as: What kind of optimization algorithm did you use? Which state estimator did you use? How did you learn the full-body dynamics? What kind of contact model did you use? This is because multiple techniques need to be properly integrated to generate the whole-body robot's motion with MPC, but these details are not usually introduced in previous literatures. Therefore, here, we would like to present several technical lessons that we learned through our experience in whole-body motion generation. Moreover, a shortcoming and a future step of the MPC approach are discussed in the end of this retrospective.

## Lessons learned from whole-body motion generation

### Real-time optimal control.
As the optimization algorithm, we used iterative Linear Quadratic Regulator (iLQR) [3]. In order to perform real-time optimal control, we used several tricks, for instance, a regularization scheme and a line search approach described in [4] to speed-up the MPC computation. The iLQR is a gradient-based optimization algorithm, and the most computationally intensive part of the algorithm is to compute the derivatives of the dynamics. Thus, a parallel computation with multiple CPU cores is a highly effective and easy way to reduce the computational time of derivatives calculation.

### State estimation.
The control performances of a controller greatly depend on estimation accuracy. On the other hand, a fast estimation is another crucial issue for real-time MPC. To perform the fast and accurate state estimation, we initially tried to use an extended Kalman filter and an unscented Kalman filter [5] as whole-body state estimators. These estimation methods worked quite well if the humanoid robot moved slowly. However, we sometimes observed unstable estimation results for the base position and the velocity due to the highly nonlinear nature of the full-body humanoid dynamics. An alternative practical way is to design a state

estimator for the base link independently by decoupling the dynamics of the base link from the full-body dynamics [6]. In this formulation, the base dynamics can be modeled as a linear system. We eventually decided to design the base state estimator using a particle filter to integrate the information of an inertial measurement unit and a laser rangefinder.

**System identification.**
The knowledge of the inertial parameters of a humanoid robot is crucial for MPC. In order to construct a realistic simulation environment for the optimization, we have to obtain the accurate parameters of the full-body dynamics such as the mass, the CoM position, and the inertia matrix of each body segment. They can be provided by a Computer Aided Design (CAD) data. However, we found that they are not reliable because they do not take some parts of actuators, cabling or additional covers into account. We thus estimated the inertial parameters using a real robot's motion data.

There are two formulations for the system identification: learning a forward dynamics model or an inverse dynamics model. We currently use an approach to learn the forward dynamics model [7] because the forward model is utilized in MPC. However, we initially tried to use the formulation for the inverse model since the inverse dynamics model can be learned efficiently for high-dimensional systems [8]. What we found is that the estimated parameters with the inverse dynamics model are not always accurate for forward model since the humanoid robot is a highly nonlinear system. We also tried to re-optimize the estimated parameters using the learning approach for the forward model. The result was not so different when they were learned from scratch in the forward dynamics model.

**Contact modeling.**
The contact model could significantly affect to the optimization results in terms of control performance and computational time. A realistic behavior can be yielded with a contact model where an optimization problem is solved (e.g. [9]), but it takes much computational time for the optimization. On the other hand, the contact impulse can be computed very fast by using a contact model with a nonlinear spring-damper system [10], but we had to tune the spring stiffness and damping ratio for different tasks or environments. We prefer to use a contact model which can tune the spring-dampers online [11]. However, sometimes a too-large penetration occurs, or too-small friction forces are computed with the contact model. Contact models still needs to be much improved for real robot control.

**Future step of MPC.**
One shortcoming of MPC (and also Optimal Control) is that how to design an objective function of an optimal control problem is not clear. The connection between task goals and behaviors of a many-DOF robot system is not always obvious. For example, if the task goal is to do a handstand walk, what is the objective function for a humanoid robot?

The shortcoming can be remedied by estimating the objective function from the demonstrations of experts using Inverse Optimal Control (IOC) (e.g., [12]). We verified that objective functions for squat and jump motions could be learned from captured human movements using an IOC approach [13]. Our future step includes a development of a humanoid control framework composed of our full-body MPC approach and IOC to effortlessly generate a wide variety of agile motions.

## Acknowledgments

This work was supported by JSPS KAKENHI under Grant JP16H06565. Part of this research was supported by MIC-SCOPE (182107105), NEDO, JST-Mirai Program Grant Number JPMJMI18B8, Japan.

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
