# OpenReview forum: "MPC for humanoid control"
_roboticsfoundation.org/RSS/2020/Workshop/RobRetro — RobRetro 2020_

### Official Review · AnonReviewer1 · 2020-06-23
**Interesting retrospective on humanoid control, could have some more details**

**Confidence:** 5
**Rating:** 8

**Review:**

This paper presents an interesting retrospective on full body humanoid control. The authors introduce a recent paper on MPC for full body control, and the details of the control that made the hardware demonstration possible. The provided insights in state estimation, system identification and contact modeling are interesting, and helpful for new researchers in full-body humanoids.

The submission could benefit with more details on the MPC algorithm itself, and tricks and insights that went into making the control real-time. The authors should certainly cite their submitted work, so the readers can have better context. Some additional questions that come to mind:
1. How did the authors speed up the full-body optimization? A very brief overview of the paper would be good.
2. How important was the full-body MPC optimization vs inverted-pendulum+inverse-dynamics (QPs) control for this problem?
3. Why did the authors choose skating vs walking as the problem to demonstrate their algorithm?
4. What could be interesting future steps for this algorithm? What can authors achieve with full-body MPC now that couldn't be done before?
5. What are the shortcomings? Contact point assumption? No slipping?

Some follow-up clarifications from the paper:
1. System identification: what was the chosen approach for the experiments in the paper?

There are also a few grammatical errors in the paper:
1. 'environments as extremely' -> environments such as extremely
2. 'they remain difficult to generate human-like behaviors' -> they still have difficulty in generating
3. 'An altanative practical' -> An alternative practical

Overall, an interesting and very useful retrospective. The paper would benefit with more details on the implementation of the MPC algorithm, which will also make it more useful for beginners in MPC and humanoid control in the robotics community.

---

### Decision · Program_Chairs · 2020-06-25

Accept